# Toxocariasis as a Rare Parasitic Complication of a Transthoracic Spine Surgery Procedure

**DOI:** 10.3390/medicina57121328

**Published:** 2021-12-03

**Authors:** Jan Soukup, Jan Cerny, Martin Cegan, Petr Kelbich, Tomas Novotny

**Affiliations:** 1Department of Orthopaedics, University J.E. Purkinje, 401 13 Usti nad Labem, Czech Republic; soukup07@kzcr.eu (J.S.); jan.cerny2@kzcr.eu (J.C.); 2Department of Rehabilitation and Sports Medicine, Second Faculty of Medicine, University Hospital Motol, Charles University, 150 06 Prague, Czech Republic; 3Department of Pathology, Masaryk Hospital Usti nad Labem, 401 13 Usti nad Labem, Czech Republic; martin.cegan@kzcr.eu; 4Department of Biomedicine and Laboratory Diagnostics, Masaryk Hospital, University J.E. Purkinje, 401 13 Usti nad Labem, Czech Republic; petr.kelbich@kzcr.eu; 5Department of Clinical Immunology and Allergology, Faculty of Medicine and University Hospital in Hradec Kralove, Charles University in Prague, 500 03 Hradec Kralove, Czech Republic; 6Laboratory for Cerebrospinal Fluid, Neuroimmunology, Pathology and Special Diagnostics Topelex, 190 00 Prague, Czech Republic

**Keywords:** *Toxocara*, spine surgery, thoracic approach, complication, pleural effusion

## Abstract

Human toxocariasis is a helminthozoonosis caused by the migration of *Toxocara* species larvae through an organism. The infection in humans is transmitted either by direct ingestion of the eggs of the parasite, or by consuming undercooked meat infested with *Toxocara* larvae. This parasitosis can be found worldwide, but there are significant differences in seroprevalence in different areas, depending mainly on hot climate conditions and on low social status. However, the literature estimates of seroprevalence are inconsistent. Infected patients commonly present a range of symptoms, e.g., abdominal pain, decreased appetite, restlessness, fever, and coughing. This manuscript presents a case report of a polytraumatic patient who underwent a two-phase spinal procedure for a thoracolumbar fracture. After the second procedure, which was a vertebral body replacement via thoracotomy, the patient developed a pathologic pleural effusion. A microscopic cytology examination of this effusion revealed the presence of *Toxocara* species larvae. Although the patient presented no specific clinical symptoms, and the serological exams (Enzyme–linked immunosorbent assay (ELISA), Western blot) were negative, the microscopic evaluation enabled a timely diagnosis. The patient was successfully treated with albendazole, with no permanent sequelae of the infection.

## 1. Introduction

Human toxocariasis is a helminthozoonosis caused by the migration of *Toxocara* species larvae through the organism. It was first described in 1952 and is distributed worldwide, with several endemic regions. The seroprevalence ranges from 4.2% up to 65.4% [1,2]. In the Czech Republic, the seropositivity rates range from 1.4% to 7.5% and various studies suggest that there has been a progressive decrease in *Toxocara* seroprevalence in the Czech population [3]. Humans can be infected either by direct ingestion of the eggs of the parasite or by consuming the larvae in undercooked meat. Toxocariasis is the most common zoonotic helminthic infection in industrialized countries. However, apart from some cases mentioning neuromuscular complications of toxocariasis [4,5], there is no known literature source that discusses the onset of this type of infection as a complication of a spinal procedure. This manuscript presents the case of a patient who suffered polytrauma after a suicide attempt, the most serious injury being a complex spinal fracture requiring a two-phase procedure. The second spine surgery was complicated by the development of a massive thoracic effusion, in which we revealed the presence of *Toxocara* species larvae. Although patients with this kind of infection commonly show a range of symptoms (e.g., abdominal pain, decreased appetite, restlessness, fever, coughing, etc.), our patient was asymptomatic, and the serological exams (Enzyme–linked immunosorbent assay (ELISA), Western blot) were negative. Therefore, the diagnosis was dependent solely on microscopic examination. Nevertheless, the diagnosis was timely, and thanks to effective treatment with albendazole, we eliminated the cause of the thoracic effusion effectively, and the patient had no permanent sequelae of the infection. We hope that this case report emphasizes the importance of vigorous alertness towards any postoperative complication, and that it will inspire clinicians worldwide to consider the possible occurrence of rare pathogens outside of the regions where they are endemic.

## 2. Case History

At admission, our patient was a 28-year-old Caucasian female. She was admitted through the emergency department of the Masaryk Hospital in Usti nad Labem as a polytrauma case of a suicidal attempt by falling from a height of 6 m. She had a history of substance abuse and chronic type C hepatitis. The clinical exam and a head-to-pelvis spiral CT revealed a flexion distraction burst fracture (Arbeitsgemeinschaft für Osteosynthesefragen (AO) types: B2 + A4) of the L1 vertebra, a transversal Roy-Camille type 1 fracture of the sacrum (S2–S3 level), compression (AO A1) fractures of the T8 and T9 vertebrae, and a comminuted fracture of the sternum (Figure 1).

Next, an X-ray of both ankles showed bilateral comminuted pilon fractures. The L1 burst fracture was evaluated as the most urgent injury due to a 70% spinal canal stenosis (the prominence of the L1 fragments of 9 mm left a residual lumen of only 4 mm). Clinically, severe paraparesis was present. Considering the circumstances, a spinal surgeon indicated an urgent L1 fracture procedure. The surgery was initiated 1.5 h after the CT scan was taken. A posterior approach was used to reach the spine. Dural sac decompression was performed through a laminectomy, followed by insertion of transpedicular screws in T12 and L2, and an L1 ligamentotaxy-mediated vertebral body reposition. The whole procedure was finished with a firm fixation of the instrumentation (Figure 2).

Taking into consideration the greater blood loss, the operation was terminated without performing posterior spondylodesis (respecting the damage control surgery principles). Under the same anesthesia, the trauma surgeons performed a bilateral external fixation of the pilon fractures. The pelvic and sternal fractures were indicated for conservative treatment. On the first postoperative day, there was a significant motor function improvement in the lower limbs (the patient was capable of active hip and knee flexion on both sides). On the third postoperative day, extraction of the external fixators and a subsequent open reduction with a plate osteosynthesis of both pilon fractures was performed. The neurological deficit continued to improve gradually. Wound healing was uncomplicated. At the 2-month post-operative follow-up, a CT scan showed the relative stability of the T12/L1 segment, and the L1 vertebral body appeared to have the potential to heal spontaneously (Figure 3). At the 6-month post-operative follow-up, we took another CT scan, which revealed no further healing tendencies. On the contrary, the defect zone in the L1 body was enlarged, and the anterior spinal column integrity was severely compromised (Figure 4). Furthermore, the scan revealed a clear vacuum phenomenon of the T12/L1 disc, which is a typical sign of segmental instability. Considering these findings, we decided that a supportive L1 vertebral body replacement was necessary to regain anterior column stability. The importance of this procedure was emphasized by the fact that, if we did not provide sufficient anterior support, the posterior instrumentation, as a stand-alone procedure, would most probably have failed in the future. The anterior procedure was not performed until 7 months after the primary injury. This long postponement was caused by the Covid-19 pandemic and the associated ban on elective procedures.

During this hiatus, the neurodeficit evolved to the stage of a mild lower limb acroparesis bilaterally (Frankel D). The patient was self-sufficient in ambulation, using two French crutches for additional support, mainly because of the tibial fractures. The anterior procedure was performed through left thoracotomy. Specifically, we partially replaced the L1 vertebral body using a distractible titanium cage, lined with autologous bone grafts (Figure 5). With regard to the sufficient decompression of the spinal canal by L1 vertebral laminectomy during the first operation and a corresponding improvement in the neurological deficit, we intentionally performed only a partial L1 corpectomy. The residual L1 dorsal prominence was therefore left intact as it did not compromise the patency of the spinal canal. No intraoperative complications were encountered. Intraoperative X-ray proved correct positioning of the implanted cage (Figure 5). A standard closure of the thoracotomy was performed, and a thoracic drain was inserted in the left anterior axillary line.

Postoperative monitoring on the orthopedic ICU was uneventful. Postoperative standing X-ray showed a slight anterior dislocation of the inferior footprint and a minor subsidence of the superior footprint of the cage into the adjacent T12 vertebral body (Figure 5). However, considering the presence of the posterior fixator after previous procedure, we evaluated the segmental stability as sufficient, regardless the change in the position of the cage. On the third postoperative day, taking into consideration the good X-ray findings (Figure 6), the drain was extracted, and the condition of the patient gradually improved. On the sixth postoperative day, the patient reported breathing difficulties, and a clinical examination found weakened breathing sounds on the left side of the thoracic cavity. We therefore took another X-ray, which showed a fluidopneumothorax on the left side, together with a pulmonary collapse (Figure 6). We consulted with a thoracic surgeon on the condition, who recommended repeating the thoracic drainage (Figure 6). Immediately after introduction, the drain aspirated 400 mL of serosanguinolent fluid.

We collected a sample of the effusion and sent it for further examination. The bacterial cultivations, including for acid-fast bacilli, were negative. Cytological energy analysis showed a relatively high number of immunocompetent cells (6 830/1 µL), with a slight predominance of monocytes and macrophages (about 40%) and a smaller number of neutrophils (about 30%), lymphocytes (about 15%), and eosinophils (about 15%), and normal energy ratios in the pleural cavity (coefficient of energy balance (KEB) = 29.17). Many foam and ring cells showed tissue damage in the pleural cavity. A flood of mixed inflammatory cells with a significant share of eosinophilic granulocytes and the presence of the *Toxocara* larvae was found by the cytological examination (Figure 7). The national reference center for microscopy evaluated the larvae of the thoracic effusion as *Toxocara canis*. An infectologist evaluated the condition as larva migrans visceralis, a stage of a parasitic infection when the larvae migrate through the internal organs.

As the patient had not presented any earlier signs of an infectious or respiratory illness, this was an accidental finding associated with a complication of the postoperative condition. We consider the quick onset of the fluidopneumothorax, associated with a pulmonary collapse, to be a consequence of the presence of the parasitic infection. The cytologic findings clearly showed the character of a parasitic effusion (eosinophilia and microscopic detection of the larvae). No increase in the number of eosinophilic granulocytes and no leukocytosis were found in the blood tests. The patient was then immediately transferred to the Department of Infectious Diseases, where albendazole 400 mg once-daily treatment was initiated. The medication was administered for 7 days (the patient weighed 65 kg). An ophthalmologist ruled out any intraocular parasite infection. Three stool samples were collected, and no parasites or eggs were found. Furthermore, the serological examination (ELISA) did not reveal any presence of common-type parasite antibodies (Strongyloidosis, Toxocariasis, Trichinelosis, Echinococosis, Cysticercosis, Fasciolosis, Filariosis). No liver or lung biopsy was performed because it would have posed the risk of another procedure, and the presence of the parasite was already proven in the pleural effusion. After adequate treatment of the parasitic infection, the patient was discharged home. The operation wounds were all healed without complications. The patient presented no signs of back pain. After an interval of two months, the serological examination for helminthosis was repeated, again with negative results. No more signs of pulmonary or intrathoracic pathology were encountered.

## 3. Discussion

Larva migrans visceralis syndrome is a complex of symptoms associated with larval infection of internal organs. Clinically, the disease can be very multifarious, from an asymptomatic course to a serious involvement of important organs, such as the lungs, liver, eyes, or brain [6,7]. Some helminthic infections may even lead to blindness [8,9]. Brain infection can lead to behavioral changes in the host [10]. Toxocariasis is generally more common as a childhood infection, mainly caused by ingestion of the eggs of the parasite, which can often be found in soil mixed with animal excrements. The eggs can be ingested together with contaminated vegetables, and the larvae can be ingested together with insufficiently cooked meat of an intermediate host (e.g., a chicken, a lamb, or a rabbit) [11]. According to a Czech study, which investigated the contamination of soil samples with eggs of the *Toxocara* species, it was found that the most infested places were backyards inhabited by feral cats, followed by parks and shelters [12]. Because the patient did care for any household animals, the primary cause of the infection could more likely be associated with drug abuse and with a poor living environment in general. We can only speculate on whether the suicide attempt had anything to do with the parasitic infection, but there were no signs of brain damage related to toxocariasis. The diagnosis of toxocariasis is generally based on an evaluation of clinical symptoms and laboratory tests. Although the symptoms can be nonspecific, the most common are stomachache, coughing, and respiratory distress. The laboratory diagnosis is based on the detection of parasite-specific antibodies using the ELISA method. Eosinophilia and hypergammaglobulinemia are less specific but also useful parameters [13]. It was recently found that mere evidence of an increase in the values of IgG, IgM, IgE, IgA or circulating antigens cannot ultimately identify or differentiate an active or a past infection. These values should therefore always be correlated with the laboratory and clinical findings [14]. The case reported here is remarkable: even though there were no serious clinical symptoms, and the laboratory blood parameters were physiological, the microscopic examination objectively proved the presence of parasite infection and eosinophilia in the thoracic effusion. The serological blood examination repeatedly did not reveal any presence of toxocara-specific or other helminthic antibodies. Although serology often shows cross-positivity among different helminthic species [15], its sensitivity is not 100%, and a negative result does not preclude the diagnosis [16]. Fecal examination for the presence of eggs is not entirely suitable for human toxocariasis, as the larvae do not evolve into their adult forms in a human host, and the eggs are therefore not excreted in the stool. Accordingly, three stool samples did not prove the presence of the parasite in this case. Vehement and rapid development of a thoracic effusion which would be responsible for a pulmonary collapse is not a very common consequence of a thoracotomy. We therefore considered such a finding to be a complication, and we carried on with microbiological, biochemical, cytological, and microscopic examinations of the effusion. In this particular case, the most probable causes of such a complication were either a bacterial infection or a tuberculous infection. Tumorous spreading had to be considered as well. The bacterial cultivation was negative. The finding of a small number of erythrophages and siderophages revealed a mild hemorrhage in the same area. On the other hand, normal energy ratios in the pleural cavity (KEB = 29.17) excluded very intense inflammation with an oxidative burst of professional phagocytes of bacterial etiology (including extracellular bacteria and *Mycobacterium tuberculosis*) in the pleural cavity [17,18,19]. The finding of a small number of plasmocytes indicated a mild local serous inflammatory reaction with antibody production. We initially assessed the relatively high number of eosinophils as tissue repair and regeneration. However, after finding the *Toxocara* parasite, we concluded that our findings were an eosinophilic reaction against this multicellular parasitic agent in the pleural cavity, accompanied by a serous antibody-mediated inflammatory reaction, tissue damage, a mild hemorrhage, and a clean-up reaction. Finally, the microscopic evaluation of the effusion objectively revealed the etiology, although the patient had shown no symptoms of a parasitic infection. Moreover, commonly present signs of parasitic infections, such as eosinophilia in blood tests or helminth-specific antibodies, were not found. The anterior approach to the thoracolumbar junction requires a thoracotomy, sometimes supplemented by a diaphragm release (only if needed for adequate exposure of the spine). This approach can therefore be associated with a number of possible post-operative complications [20,21,22], which, in spinal surgery, generally have a prevalence of approximately 30% [23]. The most common complications are pulmonary atelectasis, pneumonia, spinal cord ischemia, cardiovascular complications, chylothorax, damage to major blood vessels and the urethra, or retrograde ejaculation as the consequence of sympathetic plexus injury [21]. However, there is no known literature source that discusses the emergence of parasitic infections after spinal surgery or thoracotomy and corpectomy. Although the onset of this parasitic infection most probably was not directly associated with the course of our spinal procedures, we were able to detect the cause of this complication relatively quickly. In addition, the diagnosis was reached even though there were no specific clinical symptoms and the serological exams were negative. This was therefore a relatively rare type of case.

## 4. Conclusions

The conclusion of our case report is that although the European region has one of the lowest seroprevalences of toxocariasis in the world [24,25], we should nevertheless, as in regions where it is endemic, always consider the potential risk of a parasitic infection, even in developed countries with a low *Toxocara* seroprevalence.

## Figures and Tables

**Figure 1 medicina-57-01328-f001:**
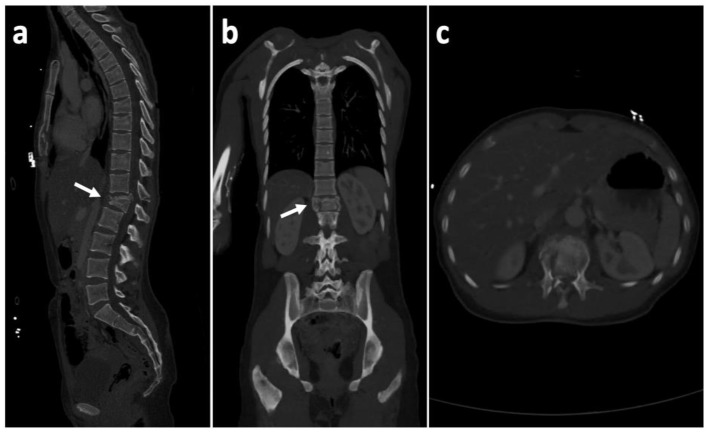
Pre-operative CT scan of the thoracolumbar junction showing an unstable L1 complete burst fracture (Arbeitsgemeinschaft für Osteosynthesefragen (AO) types: B2 + A4) (white arrow). (**a**) Sagittal section; (**b**) coronal section; (**c**) axial section.

**Figure 2 medicina-57-01328-f002:**
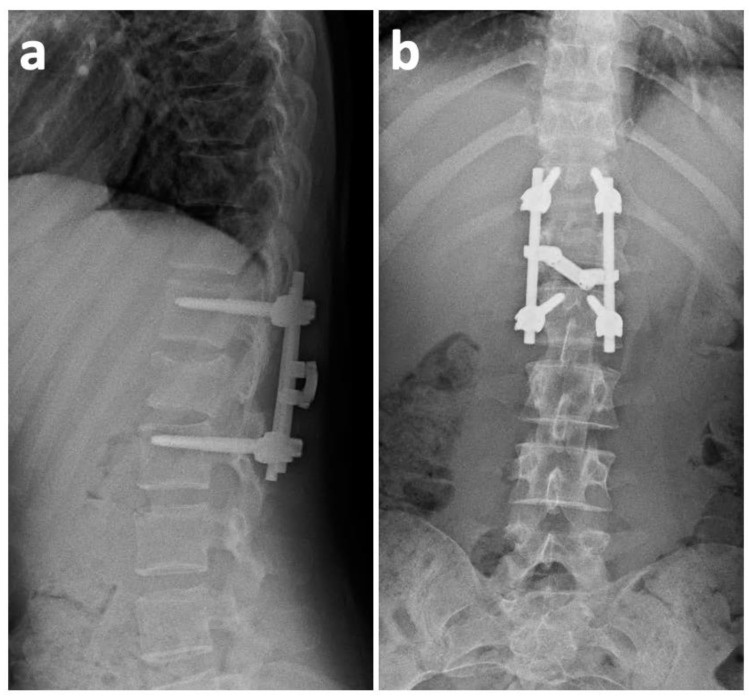
Post-operative X-ray showing optimal alignment of the transpedicular T12 and L2 screws with firm fixation of the instrumentation. (**a**) Lateral radiograph; (**b**) anteroposterior radiograph.

**Figure 3 medicina-57-01328-f003:**
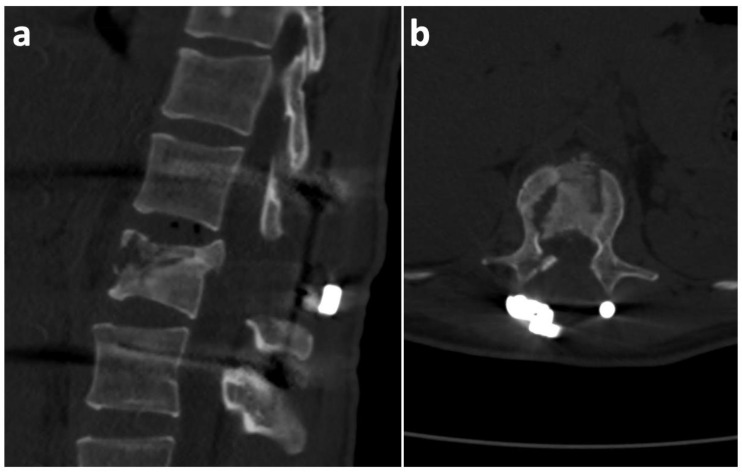
Two-month post-operative CT (sagittal and axial projections) scan at the level of the L1 body, showing good correction with the potential to heal spontaneously. (**a**) Sagittal section; (**b**) axial section.

**Figure 4 medicina-57-01328-f004:**
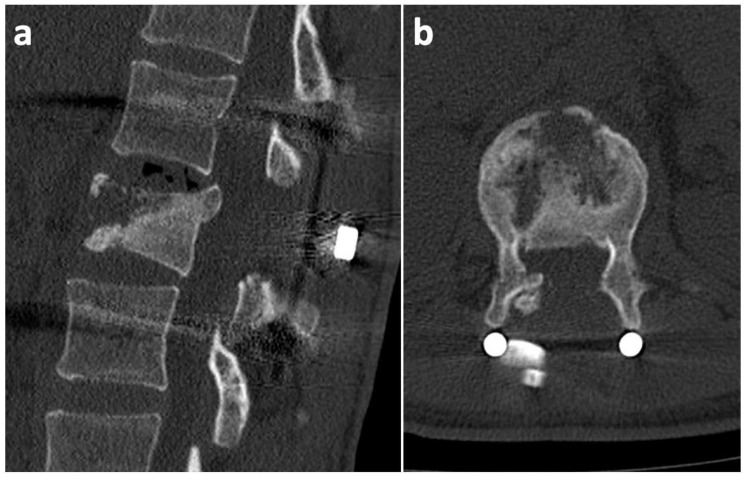
Six-month post-operative CT scan at the level of the L1 body shows worsening of the vertebral defect zone, together with a vacuum phenomenon of the T12/L1 disc, which is a clear sign of segmental instability. (**a**) Sagittal section; (**b**) axial section.

**Figure 5 medicina-57-01328-f005:**
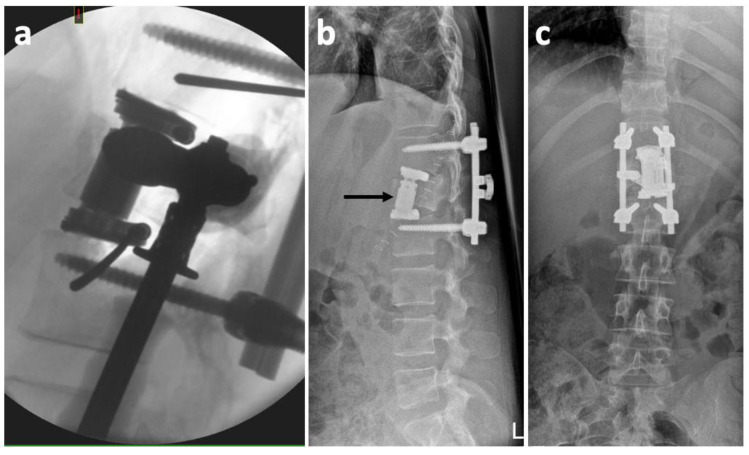
X-rays showing the partial L1 vertebral body replacement using a distractible titanium cage. (**a**) Intraoperative lateral radiograph showing correct position of the cage; (**b**) Postoperative lateral radiograph with a slight dislocation of the cage (black arrow); (**c**) Postoperative anteroposterior radiograph.

**Figure 6 medicina-57-01328-f006:**
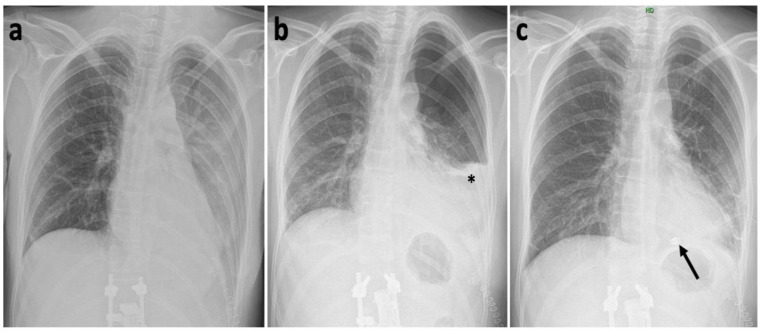
Posteroanterior thoracic X-ray (**a**) Day 3 after surgery with no pleural effusion; (**b**) Day 6 after surgery, with the newly developed pleural effusion visible (black asterisk); (**c**) Day 6 after surgery after chest drain insertion (black arrow).

**Figure 7 medicina-57-01328-f007:**
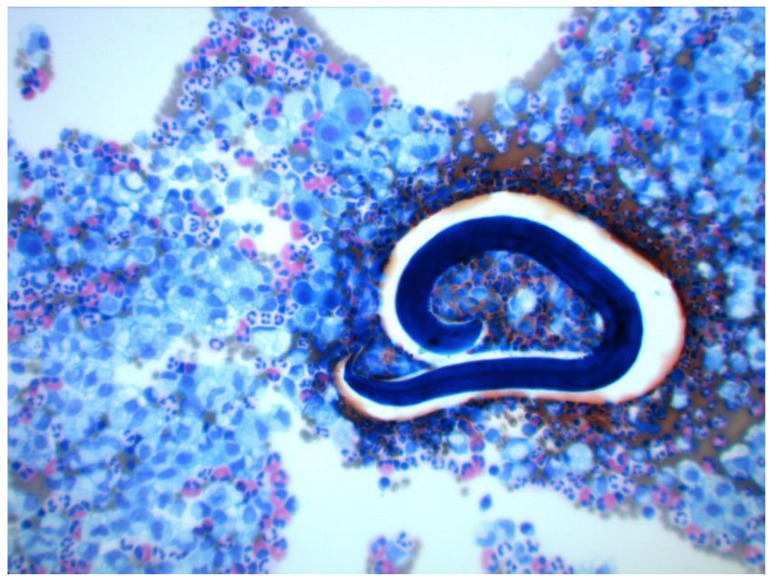
Microscopic evaluation of *Toxocara canis* larvae in the thoracic effusion.

## Data Availability

The data that support the findings of this study are available from the corresponding author upon reasonable request.

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
