# Peer review of "Toxocariasis as a Rare Parasitic Complication of a Transthoracic Spine Surgery Procedure"

_medicina, 2021, doi:10.3390/medicina57121328_

Round 1

Reviewer 1 Report

I read this interesting case report titled “Rare parasitic complication of a transthoracic spine surgery procedure,” The authors described a postoperative Toxocariasis infection after thoracotomy for spine fracture in a 28-year-old female. The T. Canis Larva was isolated from the body fluid in the thoracic cavity with no other systemic toxocariasis or serological confirmation.

Few issues were outlined as the following:

  • The title should have included the name of the patristic disease, Toxocariasis.
  • The abstract requires a revision with a paragraph explaining this rare disease and how common, Followed by a brief description of the case without surgical details. Finally, the authors need to mention there are no systemic symptoms and negative serological exams.
  • The introduction is not with the exact scope of the case report. No need to mention the surgical techniques for spine fracture. It should be deleted from page 1, lines 35 to 50, including references 1- 11. Instead, the authors should mention information on this parasitic infection, Toxocariasis, from the literature and previous neurological spine involvement (1-3). Also, it should include the geographical history of this infection, particularly in their country and the region.
  • The case history needs to be shortened. They should include the images of the fracture before and after surgery, particularly the CT image showing pleural effusion. Figure 1 revealed the T. Canis larva and not the egg, which should be mentioned. The author should specifically mention CBC and findings of eosinophile and whether they did ELISA test for T. canis antibodies or not and whether they did liver or lung biopsy or not, which may be the source of the larva.
  • The authors should discuss how common or endemic is Toxocariasis in their region as compared to other areas of the world. No need to mention the approaches of spine surgery; lines 176-182 on page 4 need to be deleted.
  • The authors should conclude from their case report that the parasitic infection should be kept in mind in the endemic region.

Selected references:

  1. Chieffi PP, Zevallos Lescano SA, Rodrigues E Fonseca G, Dos Santos SV. Human Toxocariasis: 2010 to 2020 Contributions from Brazilian Researchers. Res Rep Trop Med. 2021 May 19;12:81-91. doi: 10.2147/RRTM.S274733. PMID: 34040480; PMCID: PMC8141392.
  2. Mitsuhashi Y, Naitou K, Yamauchi S, Naruse H, Matsuoka Y, Nakamura-Uchiyama F, Hiromatsu K. [A case of the myelitis due to Toxocara canis infection complicated with cervical spondylosis]. No Shinkei Geka. 2006 Nov;34(11):1149-54. Japanese. PMID: 17087270.
  3. Bohm A, Salmon-Rousseau A, Gentil A, Dalle F. Myelitis and tenosynovitis attributed to Toxocariasis. Joint Bone Spine. 2019 May;86(3):405-406. doi: 10.1016/j.jbspin.2018.09.014. Epub 2018 Sep 28. PMID: 30273660.

Reviewer 2 Report

Authors present a case report of a patient who has undergone a two-phase spinal surgical procedure for a thoracolumbar fracture. Firstly a posterior reposition, decompression and subsequent stabilization using transpedicular screws were performed and elective anterior corpectomy after 6 months using thoracotomy.  Patient experienced pleural effusion
which had to be decompressed with another thoracic drainage and citology revealed the presence of Toxocara parasite. The patient was successfully treated with Albendazole 400 mg administered once a day for a period of seven days. 

Authors present a rare case of parasitic pleural effusion following thoracotomy for corpectomy. Although the case is described in detail, most important findings - preoperative MRI, CT following first and second surgery as well as CT of the thorax are missing. This could have shed additional light to the fact that the second surgery was postponed for 6 months - we understand that this was, as authors explained, in the face of the COVID-19 pandemic; however it is unclear which kind of traumatic spine surgery can wait so long to be performed (either the surgery is not necessary or the patient developed instability in the course of time and then requiered corpectomy). 

Patient did have an unusual clinical picture with virtualy no symptoms. However, cytological analysis of the pleural effusion in the microbiology department is (should be) a golden standard. The added value of this case report is not present. Discussion should have been enriched with a literature review on post-thoracotomy complications for corpectomy. 

Round 2

Reviewer 2 Report

The authors have made significant changes to their manuscript which can now be considered for re-evaluation.

Several issues to be considered:

-Please include if applicable MRI of the spine, especially following the last surgery - the expandable cage is located too ventrally and there is still a spinal canal stenosis ;

-Please include a literature review on parasitic complications of the spine surgery  in the past years

Round 3

Reviewer 2 Report

Authors have answered sufficiently to the remarks.

This manuscript is a resubmission of an earlier submission. The following is a list of the peer review reports and author responses from that submission.